# Identification by Reverse Vaccinology of Three Virulence Factors in *Burkholderia cenocepacia* That May Represent Ideal Vaccine Antigens

**DOI:** 10.3390/vaccines11061039

**Published:** 2023-05-30

**Authors:** Samuele Irudal, Viola Camilla Scoffone, Gabriele Trespidi, Giulia Barbieri, Maura D’Amato, Simona Viglio, Mariagrazia Pizza, Maria Scarselli, Giovanna Riccardi, Silvia Buroni

**Affiliations:** 1Department of Biology and Biotechnology “Lazzaro Spallanzani”, University of Pavia, 27100 Pavia, Italy; samuele.irudal@iusspavia.it (S.I.); viola.scoffone@unipv.it (V.C.S.); gabriele.trespidi01@universitadipavia.it (G.T.); giulia.barbieri@unipv.it (G.B.); giovanna.riccardi@unipv.it (G.R.); 2Department of Molecular Medicine, University of Pavia, 27100 Pavia, Italy; maura.damato01@universitadipavia.it (M.D.); simona.viglio@unipv.it (S.V.); 3Imperial College South Kensington Campus, London SW7 2AZ, UK; m.pizza@imperial.ac.uk; 4GlaxoSmithKline, 53100 Siena, Italy; maria.x.scarselli@gsk.com

**Keywords:** *Burkholderia cepacia* complex, reverse vaccinology

## Abstract

The *Burkholderia cepacia* complex comprises environmental and clinical Gram-negative bacteria that infect particularly debilitated people, such as those with cystic fibrosis. Their high level of antibiotic resistance makes empirical treatments often ineffective, increasing the risk of worst outcomes and the diffusion of multi-drug resistance. However, the discovery of new antibiotics is not trivial, so an alternative can be the use of vaccination. Here, the reverse vaccinology approach has been used to identify antigen candidates, obtaining a short-list of 24 proteins. The localization and different aspects of virulence were investigated for three of them—BCAL1524, BCAM0949, and BCAS0335. The three antigens were localized in the outer membrane vesicles confirming that they are surface exposed. We showed that BCAL1524, a collagen-like protein, promotes bacteria auto-aggregation and plays an important role in virulence, in the *Galleria mellonella* model. BCAM0949, an extracellular lipase, mediates piperacillin resistance, biofilm formation in Luria Bertani and artificial sputum medium, rhamnolipid production, and swimming motility; its predicted lipolytic activity was also experimentally confirmed. BCAS0335, a trimeric adhesin, promotes minocycline resistance, biofilm organization in LB, and virulence in *G. mellonella*. Their important role in virulence necessitates further investigations to shed light on the usefulness of these proteins as antigen candidates.

## 1. Introduction

*Burkholderia cepacia* complex (Bcc) is a group of Gram-negative aerobic bacteria comprising 24 different species, isolated from both environmental and clinical settings [1,2]. While they can promote plant growth and provide protection from phytopathogens [3,4], they pose an important threat not only to people affected by cystic fibrosis (CF) and chronic granulomatous disease (CGD) [5], but even to patients exposed to other risk factors, such as diabetes or renal affections [6,7]. When lungs are involved, recurrent infections can lead to a progressive loss of functions, which will eventually produce episodes of pulmonary exacerbation [8], or even a diffuse systemic infection associated with a fatal and rapid necrotizing pneumonia, called “cepacia syndrome” [9]. A growing concern is represented by an increasing number of outbreaks caused by contaminated disinfectant solutions and medical devices, leading to large and multi-structure breakouts [10,11,12]. Choosing a proper antibiotic treatment is usually difficult and relies on circumstances, as empirical treatments often fail thus leading to unfavorable outcomes and eventually to the rise of further resistant strains [13]. In this light, the need for novel strategies to control these bacteria becomes mandatory, moving the attention from infection treatment to infection prevention [11,14].

Prophylactic vaccination appears to be a valuable strategy to protect fragile populations and minimize both transmission and antibiotic resistance spread, providing an alternative to expensive and burdensome drug treatments. Today, there is no vaccine available against Bcc [15], but many groups have focused on this issue, e.g., on live attenuated formulations. A *B. cenocepacia tonB* mutant was able to induce protection from acute and lethal infection in mice, with a survival of 87.5% after 6 days [16]. Conversely, higher efforts have been undertaken toward subunit vaccines, as multiple membrane proteins could be identified using bioinformatics tools and genome-available information. For example, a mixture of OMPs and a 17 kDa OmpA-like protein complexed with mucosal adjuvant were able to induce mucosal protection against *Burkholderia multivorans* and *B. cenocepacia*, while in some cases a balanced Th1/Th2 response was observed [17,18]. Furthermore, purified OmpA-like protein BCAL2958 was able to react to serum from CF patients infected with Bcc strains, leading to IgG production and an increase in neutrophils activation markers [19]. Finally, OmpW and linocin-immunized mice were able to induce both antigen-specific responses and cytokines production, conferring protection against *B. cenocepacia* and *B. multivorans* [20]. 

Regarding antigens’ discovery, a combination of protein prediction tool and surface-protein shaving approach led to the identification of 16 extracellular proteins [21]; 3 proteins reacted to serum from Bcc-infected CF patients, thus confirming their immunogenicity. BCAL2645, one of the three identified proteins, was later proven to be involved in CFBE41o- cells adhesion and invasion [22]. Similarly, Sousa et al., [23] identified nine extracellular proteins conserved inside Bcc by reacting culture supernatant to serum from chronically infected CF patients. 

In another study, a peptidoglycan-associated lipoprotein deletion mutant was produced in *B. cenocepacia* [24] and showed a reduced mortality in *Galleria mellonella*, decreased adhesion to epithelial cells, sensitivity to polimyxin and lower proinflammatory stimulation compared to the WT, all traits making it a valuable therapeutic target. 

Proteins belonging to trimeric autotransporter adhesin [TAA] category were characterized by Pimenta et al. [25] who proved the role of BCAM2418 in adhesion to mucins and host cells. An antibody generated against its N-terminal was able to prevent this interaction and to successfully protect *G. mellonella* larvae after *B. cenocepacia* K56-2 challenge. In addition, Mil-Homens et al. [26] characterized the TAA BCAM0224 as involved in biofilm formation and swarming motility, immunoevasion and interaction with epithelial cells.

In this article, we applied, for the first time, the “reverse vaccinology” approach [27] to *B. cenocepacia* to identify novel antigen candidates. Applying prediction tools and strains conservation analysis, we identified 24 new antigens and selected 2 of them for further analysis. 

## 2. Materials and Methods

### 2.1. Bioinformatic Analysis

The complete genomes of the strains listed in Table 1 were annotated using the Rapid Annotation Subsystem Technology (RAST) tool [28]. The *B*. *cenocepacia* J2315 genome was compared with each analyzed genome and with the annotated genome of *Escherichia coli* K12. 

Proteins were selected according to the following parameters: conservation among the compared *Burkholderia* strains had to be >65%; conservation into the non-pathogenic *E. coli* K12 strain had to be <30% to prioritize Bcc-specific proteins; length of the amino acid sequence had to be >230 residues to drive the selection toward multidomain and presumably multi-function antigens; and finally, proteins with an identity of 100% in all pathogenic strains were excluded, because they did not undergo immunity selective pressure [29]. Localization prediction was performed using pSORT [30]. The functional prediction was performed using InterProScan [31]. To exclude proteins with a homology similar to human ones, a BLAST analysis was used [32]. TMHMM, Phobius and HMMTOP were employed to predict transmembrane domains and to exclude proteins with more than one transmembrane domain [33,34,35]. Proteins were analyzed using Vaxign [36] and VaxiJen [37] and the antigenicity score values were accepted when ≥90.9 or ≥0.5, respectively. Structural information, if available, was collected from RCSB PDB [38].

### 2.2. Bacterial Strains and Plasmids

The strains used in this work are listed in Appendix A. *E. coli* and *B. cenocepacia* strains were grown in Luria–Bertani (LB) broth (Difco) or in artificial sputum medium (ASM) [39], with shaking at 200 rpm, or on LB agar plates at 37 °C. When necessary, cultures were supplemented with antibiotics: tetracycline (20 μg/mL for *E. coli* and 250 μg/mL for *B. cenocepacia*), kanamycin (40 μg/mL), trimethoprim (50 μg/mL for *E. coli* and 200 μg/mL for *B. cenocepacia*), chloramphenicol (30 μg/mL for *E. coli* and 400 μg/mL for *B. cenocepacia*) and ampicillin (200 μg/mL).

### 2.3. General Molecular Biology Techniques

Upstream and downstream DNA sequences (about 500 bp each) flanking the deletion target genes were amplified using a template *B. cenocepacia* K56-2 genomic DNA. The PCR amplifications were performed using HotStar HiFidelity Polymerase kit (Qiagen, Hilden, Germany); primers sequences are listed in Appendix A. Fragments were cloned into the suicide vector pGPI-SceI-X_cm_ *Bam*HI/*Xba*I, using Gibson^®^ assembly kit (NEB, Ipswich, MA, USA). Gene deletions were performed as previously described by Hamad et al. [40]. Deletions were confirmed via PCR amplification with the primers pairs listed in Appendix A, and sequencing. The curing of the mutant strains was obtained by growth in LB medium. To complement each deleted strain, *BCAL1524* (1674 bp), *BCAM0949* (1099 bp) and *BCAS0335* (3595 bp) genes were amplified using *B. cenocepacia* K56-2 DNA as template and the primers pairs are listed in Appendix A. Fragments were cloned into the vectors pAP20 (*Eco*RI/*Xba*I) [41] and pSCrhaB2 (*Nde*I/*Xba*I) [42] using the Gibson^®^ assembly kit. The complementation plasmids were introduced into the mutants by conjugation. The growth of all the mutants was measured by culturing the strains in LB or ASM and measuring the OD600 and plating the cultures every 60 min for 24 h. Complementation experiments with the inducible vector pSCrhaB2 were carried out with 0.01% of rhamnose. 

### 2.4. Proteomic Analysis 

*B. cenocepacia* mutant strains were grown in 50 mL of LB medium or ASM for 18 h. Cells were harvested and pellets were used for membrane extraction, while supernatants were used for outer membrane vesicles (OMVs) purification. Membrane fractions were prepared as previously described [43]. For OMV purification, supernatants were collected and, to completely remove the bacterial cells from the supernatant, they were filtered through a 0.45 μm cellulose membrane. The cell-free supernatants were ultrafiltrated using a centrifugal filter Amicon^®^ (Merck Millipore, Burlington, MA, USA) with 100,000 Da NMWL, until a final volume of 1 mL, removing the non-OMV-associated proteins, such as flagella [44]. 

The analysis of the membrane fractions and of the OMVs was performed as previously described, via 2D gel electrophoresis [45]. Samples were precipitated with 10:1 2.7 M trichloroacetic acid, incubated in ice for 30′ and centrifuged for 15′ at 14,000 rpm. Pellets were then resuspended in 125 µL of 8M Urea, 4% CHAPS, 0.1 M Dithioerythritol (UCD), and 0.625 µL of 0.5% bromophenol blue and 0.7 µL of IPG buffer pH 3–10 (GE-Healthcare, Chicago, IL, USA) were added. Samples were then incubated for 1 h on an isoelectric focusing strip (pH 3–10). Isoelectric focusing was performed at 20 °C according to the program described in Table 2.

Next morning, strips were equilibrated for 12 min with buffer A (6 M Urea, 2% SDS, 50 mM TRIS-HCl pH 6.8, 30% Glycerol, and 0.1 M Dithioerythritol) and then for 5 min in buffer B (6 M Urea, 2% SDS, 50 mM Tris-HCl pH 6.8, 30% Glycerol, 0.1 M Iodoacetamide, and 125 µL of 0.5% bromophenol blue). Mass weight separation (second dimension) was performed on a polyacrylamide gel formed by a 12.5% running gel and a 5% stacking gel. Briefly, 0.5% agarose was dissolved in running buffer to immobilize the strips.

Gels were digitalized using the ChemiDoc XRS system (Biorad, Hercules, CA, USA); spots were detected using PDQuest Advanced 8.0.1 program (Biorad, Hercules, CA, USA) [46]. Experiments were performed in triplicate. 

### 2.5. Antimicrobial Susceptibility Testing for Planktonic Cells

MICs were determined in triplicate according to the EUCAST broth microdilution method [47] in U-bottom 96-well microtiter plates using LB and ASM. Nalidixic acid, amikacin, aztreonam, ciprofloxacin, minocycline, piperacillin and tobramycin were obtained from Sigma-Aldrich (Merck Millipore, Burlington, MA, USA). Levofloxacin and sparfloxacin were obtained from Honeywell Fluka™ (Charlotte, NC, USA) and Meropenem from AstraZeneca (Cambridge, UK). The MIC was determined via the resazurin method [48]. 

### 2.6. Bacterial Autoaggregation Assay

Bacterial aggregation was measured according to Bhargava et al. [49], with some modifications. Bacterial cultures were grown in LB broth, at 37 °C and 200 rpm, up to the late-exponential phase. Samples were centrifuged, cells were resuspended in Phosphate Buffer Saline (PBS) (PanReac-Applichem, Darmstadt, Germany) and optical density at 600 nm was measured and adjusted to 3. Samples were incubated at room temperature for 16 h in static conditions. After incubation, 50 µL were removed from the air–liquid interface and the OD600 was measured. The aggregation was expressed as the percentage respect to the starting OD600 = 3. 

### 2.7. In Vitro Biofilm Formation Test in 96-Well Microtiter Plates

The biofilm formation of *B. cenocepacia* K56-2, mutant and complemented strains was tested using the crystal violet staining method [50]. The bacterial cells were cultured in LB or ASM O/N at 37 °C and diluted to OD600 equal to 0.05 (about 1 × 10^7^ CFU/mL). Then, 200 μL of culture were pipetted into the microtiter plate. After 4 h of incubation, the supernatant (containing nonadherent cells) was removed and 200 μL of fresh sterile medium were added to each well and incubated for an additional 20 h at 37 °C. Biofilm biomass was quantified by staining with crystal violet and taking absorbance measurements at OD595. Results were expressed as the ratio between biofilm absorbance and planktonic bacteria absorbance. 

### 2.8. Biofilm Evaluation by Confocal Laser Scanning Microscopy

*B. cenocepacia* K56-2 and the ΔBCAM0949 strains were cultured O/N in LB and diluted to an OD600 = 0.05 in the same medium. Bacterial suspension was added to the μSlide four chambered coverslip (Ibidi, Gräfelfing, Germany) for 4 h in LB, at 37 °C. The medium was removed and fresh LB medium was added. After overnight incubation, the medium was removed, and biofilms were washed twice with physiological solution and stained with Syto 9 (Invitrogen, Waltham, MA, USA) at a final concentration of 5 μM. A 63× oil immersion objective and a Leica (Wetzlar, Germany) DMi8 with 500 to 530 nm (green fluorescence representing Syto 9) emission filters were used to take five snapshots randomly at different positions in the confocal field of each chamber. The Z-slices were obtained every 0.3 microns. For visualization and processing of biofilm images, ImageJ was used. The thickness, biomass, roughness coefficient, and biofilm distribution were measured using the COMSTAT 2 software [51]. All confocal scanning laser microscopy experiments were performed three times, and standard deviations were measured.

### 2.9. Swimming Motility Assay

Motility assays were performed as previously described [52]. Briefly, 1 μL of the overnight culture was spotted in the middle of a swimming plate (agar 0.3%), allowed to dry for 30 min at room temperature and incubated for 16 h at 37 °C. Diameters of swimming halos were measured. 

### 2.10. Infection in Galleria Mellonella

Strains were grown in LB broth, at 37 °C, 200 rpm, until OD_600_ = 0.5. Each larva was then infected with 10^5^ CFU/mL in physiological solution (PS) [53], with an injection volume of 10 µL; control was performed by injecting PS. Larvae were placed in Petri dishes and kept in the dark at 30 °C. Live/dead count and health index scores [53,54] were registered after 24 h, 48 h and 72 h. For each experiment, at least 10 larvae per group were injected.

### 2.11. Lipase Activity Assay

Tributyrin agar plates (LB medium, 0.5% tributyrin, and 1.5% agar) were used for lipase activity detection [55]. Halos formed by the lipase activity around colonies were measured. 

### 2.12. Rhamnolipid Analysis

Strains were inoculated in 10 mL of LB for 48 h at 37 °C at 200 rpm. To quantify the amount of rhamnolipids in the culture supernatant, the colorimetric orcinol assay was used [56]. A volume of the supernatant (50 to 500 μL) was diluted with water to reach the volume of 500 μL. Samples were extracted twice with 2 volumes of diethyl ether. The ether fractions were evaporated and the pellets were dissolved in 100 μL of distilled water and mixed with 100 μL 1.6% orcinol and 800 μL of 60% sulfuric acid. Samples were incubated at 80 °C at 175 rpm for 30 min. After incubation, the OD421 was measured. The rhamnolipid concentration was calculated using a standard concentration curve with rhamnose standards on the assumption that 1 μg of rhamnose corresponds to 2.5 μg of rhamnolipid [57]. 

### 2.13. Statistical Methods

Analyses were performed using Prism 9.0 (GraphPad). Comparison of more than two groups were performed with the unpaired *t*-test, one-way or two-way ANOVA.

## 3. Results and Discussion

### 3.1. In Silico Identification of Antigen Candidates

The identification of antigen candidates was performed using the “reverse vaccinology” approach [27]. The scheme of the workflow is depicted in Figure 1. The genome of *B. cenocepacia* J2315 was used as reference [58], and annotated with the Rapid Annotation using Subsystem Technology (RAST) pipeline, a fully automated annotation engine for archaeal and bacterial genomes [28]. RAST identifies protein coding genes, assigns functions and gives a prediction of subsystems present in the genomes. 

The annotation of *B. cenocepacia* J2315 genome provided 7769 protein encoding genes as an output. Each genome was compared with the annotated genome of *E. coli* K-12 and with the genomes of six *B. cenocepacia* strains and eighteen Bcc strains (Table 1). The parameters to select putative antigen candidates were as follows: identity with *E. coli* K-12 proteins less or equal to 30% and identity with the proteins of *B. cenocepacia* and Bcc strains higher than 65%. Since immunogenic proteins are generally localized on the outer membrane and have more complex structures created by multiple domains, (e.g., transmembrane and extracellular domains), only proteins longer than 230 amino acids were selected. Using these parameters, a pool of 1793 proteins was obtained and analyzed for subcellular localization and putative function. The pool was reduced to 122 candidates predicted to be outer-membrane or extracellular, potentially involved in cell adhesion, virulence or with unknown function. Further analyses were performed to evaluate the transmembrane domains prediction, the homology with human proteins and the prediction of the antigenicity of the candidates as described in the “Materials and methods” section. The result is a short list of 24 antigen candidates belonging to different protein classes: 4 lipoproteins, 4 autotransporter adhesin proteins, 5 enzymes, 5 outer membrane proteins and 6 other proteins (Table 3, Figure 1). 

Among them, BCAM0949, BCAM2418, BCAS0409 and BCAS0236 were previously described as being involved in virulence and cell adhesion [25,59,60,61]. BCAS0147 was described as surface exposed [21]; BCAM1514 and BCAS0409 were localized in the outer membrane fraction in the CF niche [62].

Three antigens, the collage-like protein BCAL1524, the lipase LipA BCAM0949 and the autotransporter adhesin BCAS0335 were selected for further analysis. The rationale for the selection is described below. Collagen-like proteins, characterized by a collagen-like (CL) domain containing Gly-Xaa-Yaa amino acid repetition and organized in a triple helix-structure, are widespread in pathogenic bacteria and highly resemble human collagen [63]. Due to their structure, they can interact with different host factors promoting adhesion, inflammation and immunoreaction [64,65]. Additionally, Grund et al., [66] described the high Th2 immune response induced in mice immunized with *B. pseudomallei* collagen-like protein 8 (Bucl8) antigens, while studies on the role of these proteins in *B. cenocepacia* are still lacking. Lipases are commonly found in clinical *B. cepacia* complex isolates [67]. While evidence of their involvement in Bcc infection is still nominal [67,68], multiple studies reported their role in virulence and pathogenesis in *P. aeruginosa* [56,69,70]. Trimeric autotransporter adhesins are known to play a key role in virulence for a wide range of bacteria [71]; Pimenta et al. [25] and Mil-Homens et al. [26,72] extensively described their role in adhesion and inflammation in epidemic *B. cenocepacia* epidemic strain K56-2, thus highlighting the putative role they may play as protective antigens.

### 3.2. Construction of Deletion Mutants of the Selected Antigen Candidates

To investigate the roles of these proteins in virulence and host pathogen interaction, deletion mutants were constructed in *B. cenocepacia* K56-2, since the amino acid sequence of the corresponding proteins was identical to J2315 and this strain is most amenable to genetic manipulation. The growth of the mutants and complemented strains was comparable to that of the WT K56-2 both in LB and in ASM media (Appendix A). 

### 3.3. Analysis of Protein Localization

To demonstrate that the antigen candidates selected are localized on the bacterial surface, a proteomic analysis of Outer Membrane Vesicles (OMV) derived from K56-2 and deleted strains was performed. As shown in Figure 2, when analyzed via the 2D electrophoresis, in the OMVs of the ΔBCAL1524 isogenic mutant strain, no spots were identified at the coordinate value corresponding to the BCAL1524 protein (isoelectric point 8.55 and molecular weight 49 kDa) (Figure 2B). In the OMVs of the K56-2 wild-type strain, a spot numbered 7308 with intensity of 27,944.8 mAu was identified at these coordinates (Figure 2A,C).

The same applied to the OMV of ∆BCAM0949 and K56-2 strains electrophoresis of the OMVs of the K56-2, in which a spot numbered 4301 was identified at pI~6.4 and MW~42 kDa and intensity 3399.7 mAu, which was not present in the ∆BCAM0949 mutant OMVs (Figure 3). The MW and pI values of BCAM0949 are 38 kDa and 6.41, respectively. 

Finally, the OMVs 2D-electrophoresis of the K56-2 showed the spot number 3701 at pI 5.3 and MW 124 kDa with an intensity equal to 4156 mAu, lacking in the ΔBCAS0335 (Figure 4). The theoretical pI and MW of the BCAS0335 are 5.0 and 117 kDa, respectively. 

All these data demonstrate that the collagen-like protein BCAL1524, the lipase BCAM0949 and the trimeric-autotransporter adhesion BCAS0335 are expressed in the outer membrane vesicles compartment in the conditions tested. Two-dimensional gel electrophoresis full images are shown in Appendix A.

### 3.4. Antibiotic Susceptibility 

To test the involvement of the selected proteins in antibiotic susceptibility, the minimal inhibitory concentration of 12 currently used antibiotics was determined against the deleted and complemented strains grown in LB. Piperacillin had an MIC which was four-fold lower in the ΔBCAM0949 strain with respect to the K56-2 in LB (32 μg/mL vs. 128 μg/mL) (in bold in Table 4). The deleted strain, complemented with the WT copy of the gene cloned into the rhamnose inducible vector pSCrhaB2, showed a reverted phenotype. 

The MIC of minocycline was eight-fold higher in the ΔBCAS0335 strain than the K56-2 (64 μg/mL vs. 8 μg/mL) (in bold in Table 4). In this case, the complementation using the inducible vector pSCrhaB2 restored the phenotype of the K56-2 strain. 

The MICs of the control strains carrying the empty pSCrhaB2 vector are listed in Appendix A. It is to be noted that the presence of the empty vector did not affect strain phenotypes. 

The MICs value for the ΔBCAL1524 did not show differences compared to the K56-2 in the conditions tested.

### 3.5. Phenotypic Characterization of the Deleted Mutant

#### 3.5.1. Biofilm Formation 

To assess the role of the three antigen candidates in sessile lifestyle, biofilm formation was evaluated using crystal violet assay in 96-well plates after 48 h of incubation. The results showed that the ∆BCAM0949 mutant strain produces a lower quantity of biofilm in both LB and ASM media with respect to the WT (Figure 5A,B). The ∆BCAS0335 mutant produced a slightly lower quantity of biofilm only in LB, compared to the K56-2 (Figure 5A). Complementation was carried out using the pAP20 constitutive vector. The complemented strains showed the reversion of the ∆BCAM0949 phenotype in LB and ASM (Figure 5C,D). Regarding the ∆BCAS0335 mutant, the mutant transformed with the empty vector did not show a significant decrease in biofilm production (Appendix A), but in the deleted strain the constitutive expression of the *BCAS0335* gene significantly increased the biofilm production in both the media tested (Figure 5C,D). The data suggest that both the lipase BCAM0949 and the autotransporter adhesin protein BCAS0335 contribute to biofilm formation. On the contrary, the collagen-like protein does not seem to have a role in this process in the condition tested. The data regarding the biofilm production in the control strains of the complementation carrying the empty pAP20 vector are reported in Appendix A.

Since the difference in biofilm formation with respect to the K56-2 strain was higher in the lipase mutant, we decided to evaluate the biofilm formed by this strain using confocal laser scanning microscopy (CLSM). The obtained imaging showed that the biofilm formed by the ∆BCAM0949 mutant is less structured and thick in both conditions tested (LB and ASM) (Figure 6A). 

The analysis using COMSTAT2 confirmed the qualitative evaluation of the biofilm. Biofilms formed in LB and ASM by the ∆BCAM0949 strain have a significantly lower biomass and average thickness (Figure 6B,C). Moreover, the roughness of the biofilm is significantly higher for the deleted strain (Figure 6B,C), indicating that the structure of the biofilm is altered. 

All these data suggest that the lipase BCAM0949 has a role in biofilm formation and structure in the conditions tested. 

#### 3.5.2. Bacterial Autoaggregation

The involvement of the antigen candidates in bacterial autoaggregation, a process known to be involved in bacterial colonization and persistence in the host, was assessed using a precipitation-based assay. The decrease in the OD600 of the air–liquid interface after 16 h of static incubation is consistent with an increase in bacterial interaction, which promotes precipitation; on the contrary, if interactions are impaired, the OD600 is higher. The autoaggregation is expressed as the percentage of OD600 of the air–liquid interface with respect to the initial OD600 normalized to 3. For the K56-2 strain, the obtained value was 11.9%, while in the case of the ∆BCAL1524, ∆BCAM0949 and ∆BCAS0335 mutant strains it was 17.7%, 9.13% and 14.8% (Figure 7A), respectively. 

When BCAL1524 protein was lacking, autoaggregation decreased as the percentage of residual OD600 after 16 h was increased. The proteins BCAM0949 and BCAS0335 are not involved in autoaggregation in the conditions tested. 

The complementation with the pAP20*BCAL1524* vector restored a phenotype identical to the K56-2 (Figure 7B). 

#### 3.5.3. Swimming Motility

Swimming motility allows bacteria to diffuse through low viscosity medium, such as mucus, and it is considered a virulence determinant. Swimming motility of the deleted strains was tested on 0.3% agar plates. In the ∆BCAM0949 mutant strain, the swimming phenotype was abolished (diameter of the motility halo 10.8 mm) compared to the K56-2 (27.8 mm) (Figure 8A,B). On the other hand, the two deleted mutant strains, ∆BCAL1524 and ∆BCAS0335, showed no differences compared to the K56-2 (Figure 8A,B). 

The complementation was carried out using the entire *BCAM0949*-*BCAM0950* operon cloned into the vector pAP20 as the phenotype was not restored by expressing the only LipA (BCAM0949) protein. As reported by Papadopoulos et al. [73] and Putra et al. [74], *P. aeruginosa* LipA and *Burkholderia territori* LipBT lipases have to be co-expressed with the relative foldase to avoid precipitation and to allow increased functional protein concentration, which would be required to complement some phenotypes. Indeed, upon complementation with the pAP20*BCAM0949-0950* construct, the swimming halo was restored to the diameter of the K56-2 (Figure 8C,D). The data suggest that while the proteins BCAL1524 and BCAS0335 are not involved in swimming motility, the lipase BCAM0949 has a pivotal role in this pathway in the conditions tested.

### 3.6. Infection in Galleria mellonella

*G. mellonella* larvae are a well-known invertebrate animal model for infection, as they provide a simple, fast, and cost-effective platform. After an infection with a dose of 10^3^ CFU bacteria, symptoms started to appear 24 h post-infection (Figure 9A,B): at this time point, larvae infected with the ∆BCAL1524 and ∆BCAS0335 mutant strains showed a significantly higher health index score compared to the larvae infected with the K56-2. These larvae showed a lower melanization level than the larvae infected with the K56-2 (Figure 9B). On the contrary, infection with the ∆BCAM0949 mutant did not affect larval survival and health index scores compared to what was observed in larvae infected with the WT strain (K56-2).

To complement the observed phenotypes, the pAP20 vector was used. Complementation was achieved in the case of ∆BCAL1524 (8.15) and ∆BCAS0335 (7.9) as the K56-2 symptom score at 24 h post-infection was 7.56 (Figure 9C). The data regarding the health index score of *G. mellonella* moths infected with the control strains of the complementation carrying the empty pAP20 vector are reported in Appendix A.

The data obtained suggest that the proteins BCAL1524 and BCAS0335 are involved in virulence in vivo, since their lack increases the health score of larvae injected with the corresponding deleted strains; thus, host immunization using these two antigens could provide protection from severe infection. On the contrary, deletion of BCAM0949 does not affect the in vivo virulence of the bacterium.

### 3.7. Characterization of the Lipase BCAM0949 

#### 3.7.1. Lipolytic Activity

Although the deletion of BCAM0949 has no effect on the in vivo virulence of *Burkholderia*, its amino acid sequence is characterized by a conserved domain of the triacylglycerol esterase/lipase superfamily, so we further characterized its activity. In other bacteria, such as *Pseudomonas aeruginosa*, several lipolytic enzymes are secreted or surface-exposed; among these is the EstA, which has a lipolytic activity, and is involved in rhamnolipid production and virulence [57]. To better characterize the lipase BCAM0949, the lipolytic activity of the deleted strain was compared to the K56-2 using the tributyrin agar plate assay (Figure 10). 

The diameter of the lipolytic halo was measured and, as shown in Figure 10, the mutant strain has a lower lipolytic activity than the K56-2. The lipolytic activity is not completely blocked because the bacterium possesses other lipolytic enzymes. The complementation restored the lipolytic activity (Figure 10). 

#### 3.7.2. Rhamnolipid Production

In *P. aeruginosa*, EstA is involved in rhamnolipid production [57]. To study the involvement of the lipase BCAM0949 in rhamnolipid production, the quantity of rhamnolipids produced by the deleted strain was compared to the K56-2, showing that the deleted strain ∆BCAM0949 is impaired in their production (Figure 11). 

Complementation with the pAP20BCAM0949 vector not only restored the K56-2 phenotype, but induced an increase in rhamnolipids production (Figure 11). These results demonstrate the involvement of the protein BCAM0949 in rhamnolipid production. 

All these data showed how the protein BCAM0949 has a role in the virulence pathway of rhamnolipid production. Since rhamnolipids play a role in outer membrane composition, cell motility and biofilm formation in other bacteria, BCAM0949 is also demonstrated to be involved in different virulence pathways.

## 4. Conclusions

In this work, reverse vaccinology was used to identify surface-exposed proteins which can be ideal antigen candidates for the development of a vaccine against the *B. cepacia* complex bacteria. We chose this strategy that previously revealed to be resolutive in cases for which the discovery of a vaccine seemed to be impossible [75]. It represents therefore a valuable option in the case of *Burkholderia* for which, despite numerous attempts to find suitable antigen candidates, a vaccine is not yet available. Starting from the bacterial genome, a short list of gene candidates, encoding extracellular proteins predicted to be ideal targets for the immune system, was selected. Among the 24 proteins identified, 3 of them were selected because of their homology with important virulence factors in other bacteria: BCAL1524, a lipoprotein containing a collagen-like triple helix repeat, able to interact with host cell adhesive proteins; BCAM0949, an extracellular lipase; BCAS0335, a trimeric autotransporter adhesin (TAAs) which interacts with the extracellular membrane components and host cell receptors and possesses hemagglutinin activity. Antigens used to develop vaccines do not necessarily have to be virulence factors, however, directing the immune response toward virulence determinants has been a successful rationale in eliciting protective immunity [76]. 

Hence, the role of these proteins in virulence was investigated by constructing markerless deletion mutants of *B. cenocepacia* K56-2. 

An important aspect of antigen candidates is their localization. To support the in silico prediction, a proteomic analysis of the deleted strains was performed and it demonstrated that the three proteins are indeed localized in the outer membrane vesicle (OMV) compartments. The OMVs play a critical role in host–pathogen interaction and virulence, as they are produced in response to stress conditions and can carry bioactive molecules and virulence factors able to sustain bacterial growth and modulate host inflammation [77]; therefore, the presence of the three proteins in these compartments highlights their possible role in these pathways. Moreover, their localization on the bacterial surface could make them ideal targets for recognition by the host immune system.

The antibiotic susceptibility tests showed that BCAM0949 is involved in piperacillin resistance and BCAS0335 is involved in minocycline sensitivity. Considering the role and the localization of the two proteins, these results suggest that their absence may alter the membrane permeability, thus changing *Burkholderia* antibiotic susceptibility.

Virulence determinant analysis showed that the collagen-like protein BCAL1524 is involved in bacterial autoaggregation and has a role in *G. mellonella* infection, while the autotransporter adhesin BCAS0335 is involved in biofilm formation and promotes *G. mellonella* infection. This makes BCAL1524 and BCAS0335 suitable for further investigations as putative antigens. 

On the other hand, the extracellular lipase BCAM0949 is involved in different virulence pathways: biofilm formation, autoaggregation and swimming motility. Further investigations on this protein showed its role as lipolytic enzyme and in rhamnolipid production, highlighting its similarity with the protein EstA of *P. aeruginosa* which is involved in virulence. Thus, considering their phenotypic analogy, LipA may be a homologous candidate of EstA protein, which still has not be found in *B. cenocepacia.* Hence, these data suggest that BCAM0949 could be considered as a protein which has a pivotal role in *B. cenocepacia* virulence. 

Additional experiments will be fundamental to evaluate the immunogenicity of these antigens in vivo in the mouse model. On the basis of the potential role that these antigens play in virulence and their localization on the bacterial surface, it might be expected that the antibodies may impair their function by binding the antigens, thus generating the same phenotype of the isogenic mutant strains: reduction in adhesion, biofilm formation, membrane permeability and swimming motility.

Overall, this study provides the rationale for future functional characterization of novel vaccine candidates against Bcc, paving the way to discovering effective vaccines to prevent *Burkholderia* infections.

## Figures and Tables

**Figure 1 vaccines-11-01039-f001:**
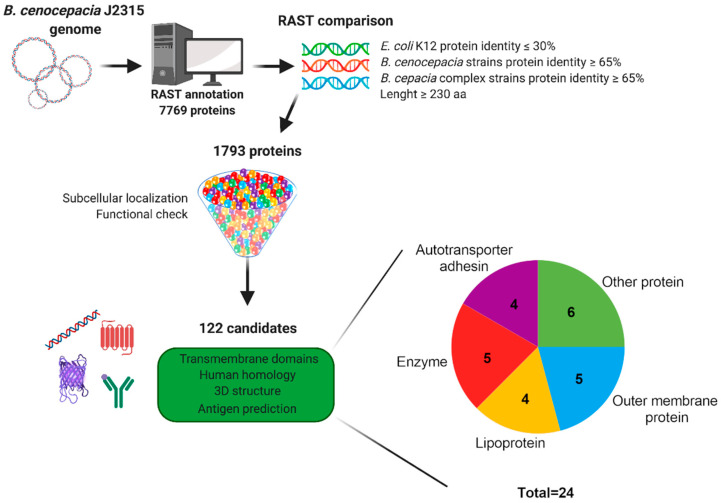
Schematic workflow of the in silico analysis. Reverse vaccinology was applied to the *B. cenocepacia* proteome to select novel vaccine candidates. The process starts with RAST software annotation and comparison, which led to the selection of 1793 candidates from 7769 proteins. These candidates were analyzed via different bioinformatics tools and bibliographic information, which further selected 122 proteins. At the end, 24 antigen candidates were selected based on the presence and number of transmembrane domains, low homology with human proteins, 3D structure and antigenicity prediction (created with BioRender.com, accessed on 5 May 2023).

**Figure 2 vaccines-11-01039-f002:**
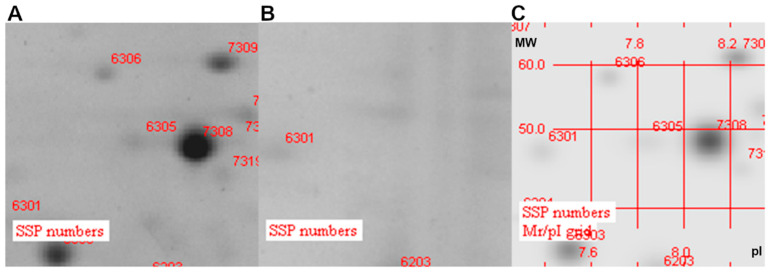
Zoom-in of two-dimensional gel electrophoresis (2DE)-stained gel with Coomassie. (**A**) 2D gel of K56-2 OMVs; (**B**) 2D gel of ∆BCAL1524 OMVs and (**C**) the 2D gels of OMV proteins superimposed with K56-2 and ∆BCAL1524. (MW: molecular weight; pI: isoelectric point).

**Figure 3 vaccines-11-01039-f003:**
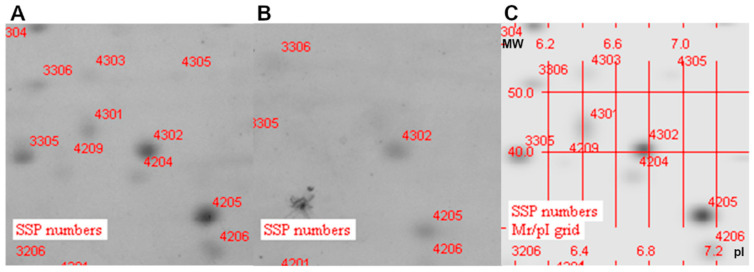
Zoom-in of two-dimensional gel electrophoresis (2DE)-stained gel with Coomassie. (**A**) 2D gel of K56-2 OMVs; (**B**) 2D gel of ∆BCAM0949 OMVs and (**C**) 2D gels of OMV proteins superimposed with K56-2 and ∆BCAM0949. (MW: molecular weight; pI: isoelectric point).

**Figure 4 vaccines-11-01039-f004:**
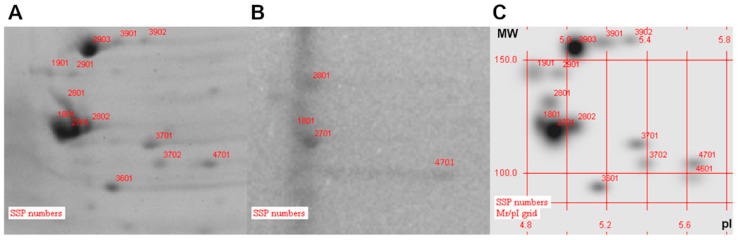
Zoom-in of two-dimensional gel electrophoresis (2DE)-stained gel with Coomassie. (**A**) 2D gel of K56-2 OMVs; (**B**) 2D gel of ∆BCAS0335 OMVs and (**C**) 2D gels of OMV proteins superimposed with K56-2 and ∆BCAS0335. (MW: molecular weight; pI: isoelectric point).

**Figure 5 vaccines-11-01039-f005:**
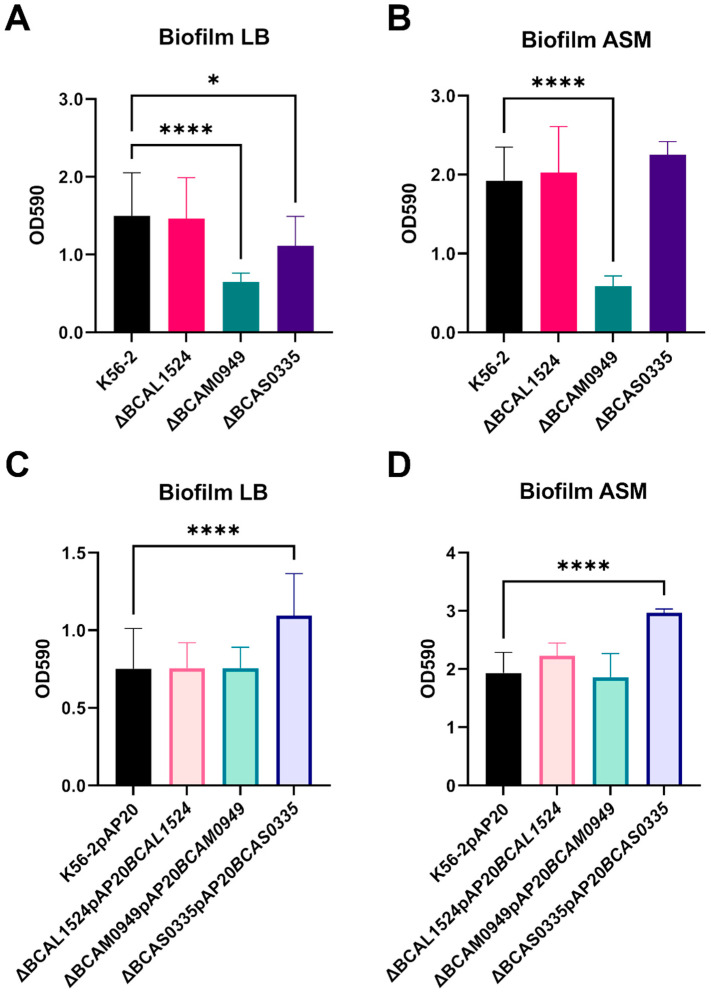
Graphical representation of the OD_590_ measured after the crystal violet assay comparing the biofilm formation of *B*. *cenocepacia* K56-2, deleted and complemented strains. (**A**) Biofilm of deleted strains in LB and (**B**) in ASM; (**C**) biofilm of complemented strains in LB and (**D**) in ASM. (* *p* < 0.1, **** *p* < 0.0001 one-way ANOVA test).

**Figure 6 vaccines-11-01039-f006:**
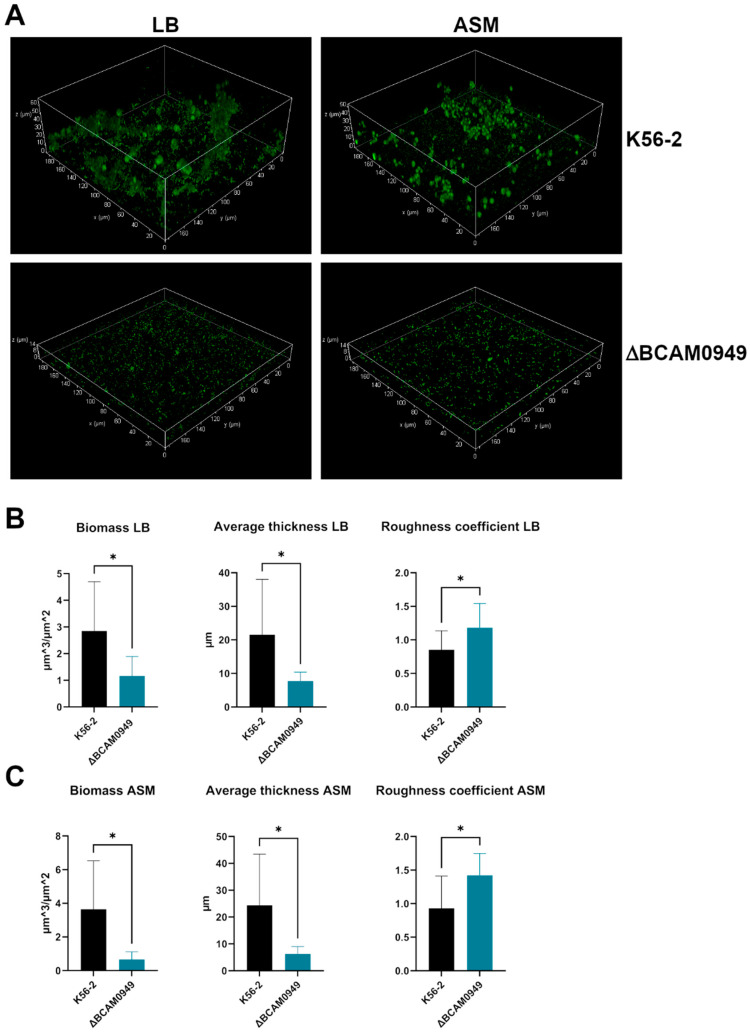
Biofilm evaluation using CLSM. (**A**) CLSM images of *B. cenocepacia* K56-2 and ∆BCAM0949 strains in LB and in ASM media. Biofilms were grown in chambered slides. Pictures were taken with an overall magnification of 400×. Cells were grown for 48 h at 37 °C in LB. Planes at equal distances along the Z-axis of the biofilms were imaged using CLSM. These images were stacked to reconstruct the 3D biofilm images. (**B**,**C**) Analysis of biofilm properties via COMSTAT 2. Measures of total biomass, average thickness and roughness coefficient are represented. Data are the mean ± SD of the results from three independent replicates. (* *p* < 0.05 unpaired *t* test).

**Figure 7 vaccines-11-01039-f007:**
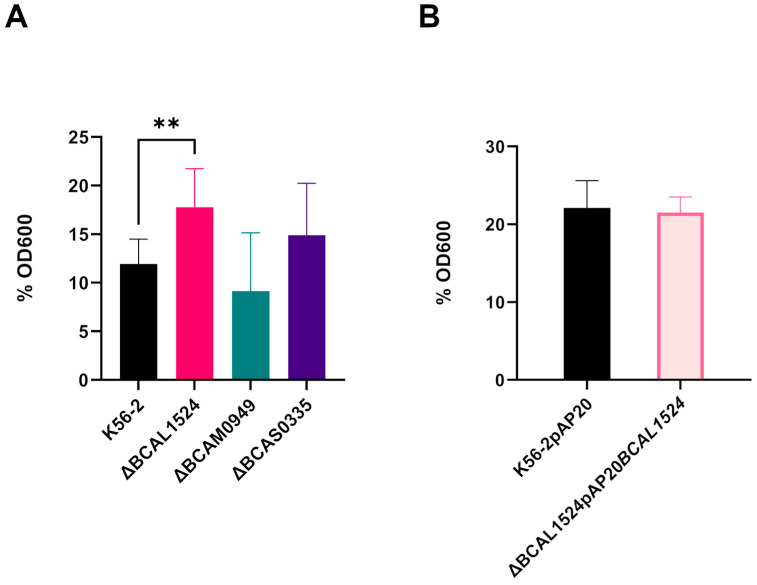
Quantitative autoaggregation assay. (**A**) Percentage of initial OD600 of K56-2 and deleted strains. (**B**) Percentage of initial OD600 of ∆BCAL1524 complemented strain. Data are the mean ± SD of the results from three independent replicates. (** *p* < 0.01 one-way ANOVA test).

**Figure 8 vaccines-11-01039-f008:**
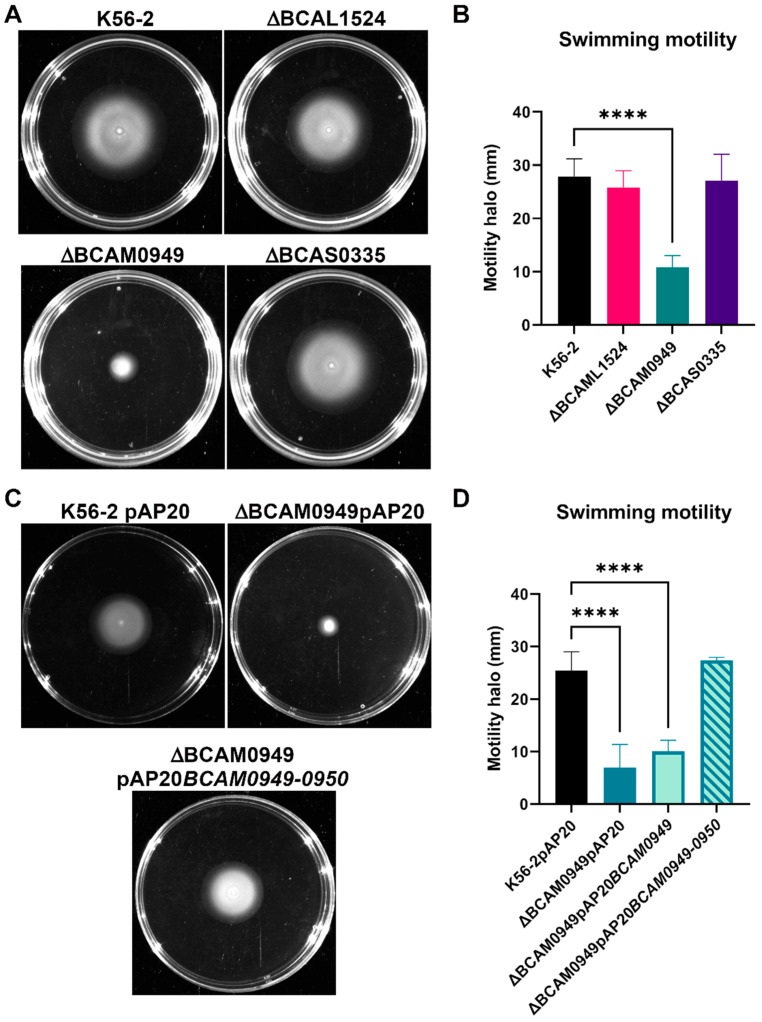
Swimming motility assay. (**A**) Swimming halo of K56-2 and deleted strains on LB 0.3% agar plate. (**B**) graphical representation of the swimming motility assay measurements (mm) of K56-2 and deleted strains. (**C**) Swimming halo of the complemented strains on LB 0.3% agar plate. (**D**) graphical representation of the swimming motility assay measurements (mm) of complemented strain. Data are the mean ± SD of the results from three independent replicates. (**** *p* < 0.0001 one-way ANOVA test).

**Figure 9 vaccines-11-01039-f009:**
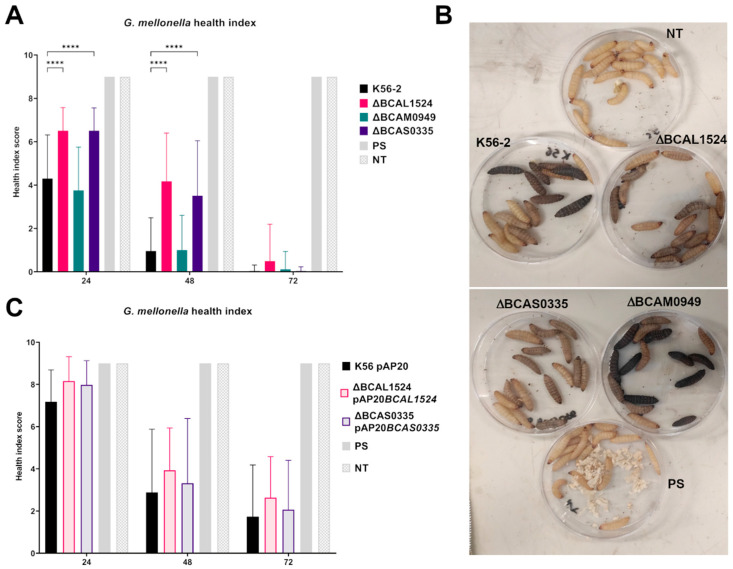
Graphical representation of *G. mellonella* health index. (**A**) Health index scores associated with infection with *B. cenocepacia* K56-2 and the mutants ΔBCAL1524, ΔBCAM0949, and ΔBCAS0335. Data are the mean ± SD of the results from three independent experimental replicates. (**** *p* < 0.0001 two-way ANOVA test). (**B**) *G. mellonella* larvae 24 h post treatment. PS: larvae injected with physiological solution; NT: not treated. (**C**) Health index scores associated with infection with *B. cenocepacia* K56-2 and complemented strains. Data are representative of the results of three independent experimental replicates.

**Figure 10 vaccines-11-01039-f010:**
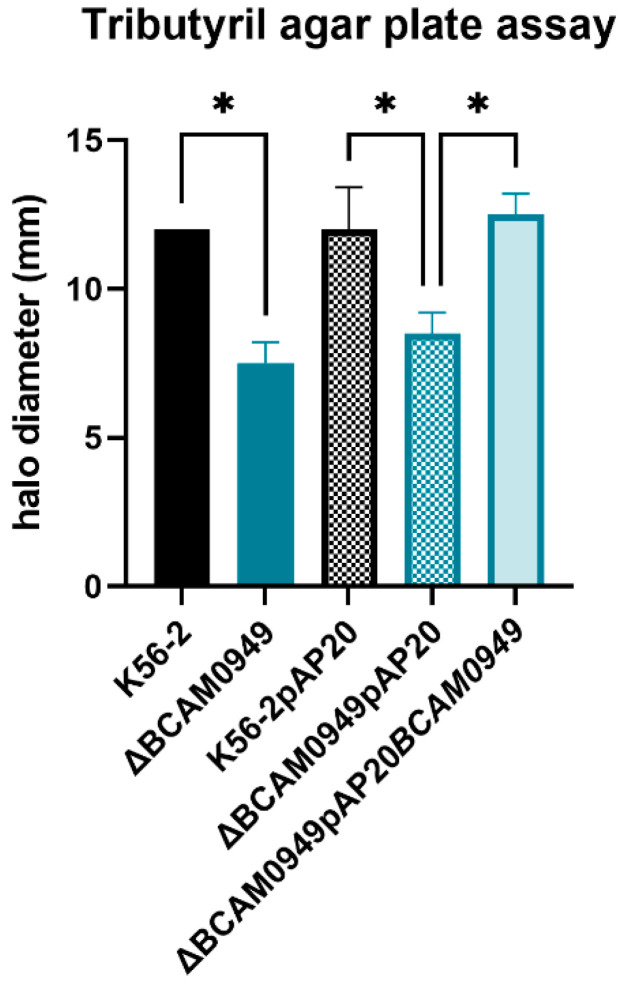
Lipolytic activity of K56-2, ∆BCAM0949 and complemented strains in terms of zone of hydrolysis (in mm) in tributyrin agar plate. Data are the mean ± SD of the results from three independent replicates (* *p* < 0.1 one-way ANOVA test).

**Figure 11 vaccines-11-01039-f011:**
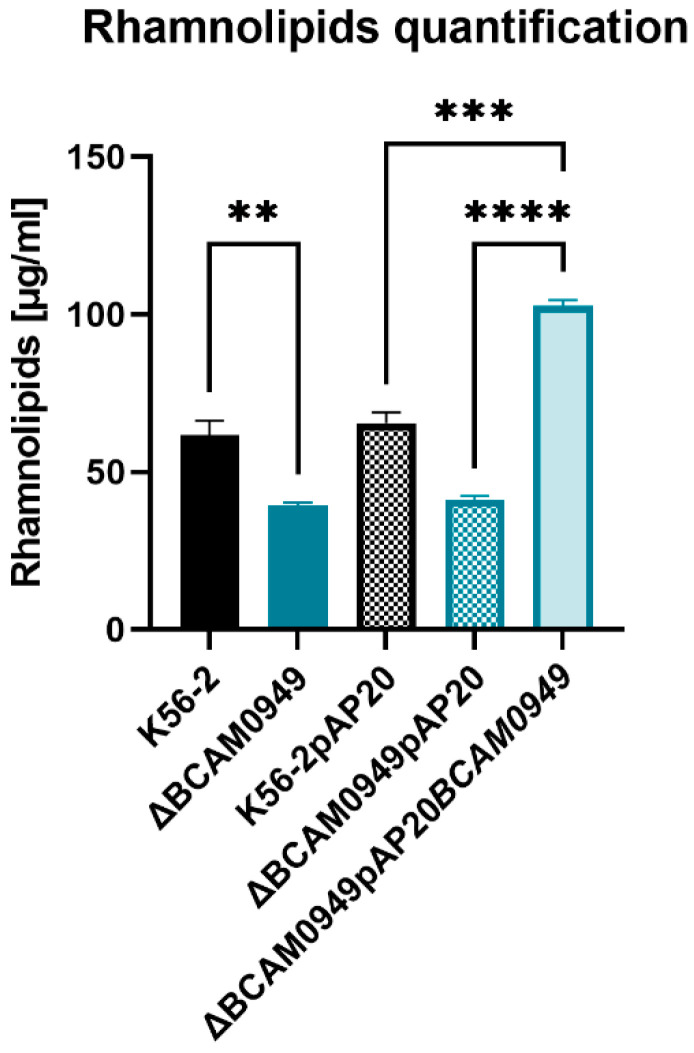
Quantitative determination of extracellular rhamnolipids produced by K56-2, ∆BCAM0949 and complemented strains as determined via the Orcinol test. SDs were calculated from three independent experiments. Data are the mean ± SD of the results from three independent replicates. (** *p* < 0.01, *** *p* < 0.001, **** *p* < 0.0001 one-way ANOVA test).

**Table 1 vaccines-11-01039-t001:** List of the *Burkholderia* genomes used for the annotation.

Strain	Source
*B. ambifaria* AMMD	cell culture
*B. anthina* AZ-4-10-S1-D7	soil
*B. cepacia* ATCC25416	wash glove
*B. cenocepacia* J2315	CF sputum
*B. cenocepacia* DDS-22E-1	aerosol sample
*B. cenocepacia* H111	CF patient
*B. cenocepacia* HI12424	cell culture
*B. cenocepacia* MSMB384WGS	water
*B. cenocepacia* VC12308	CF sputum
*B. cenocepacia* YG-3	cell culture
*B. contaminans* SK875	pig with swine respiratory disease
*B. diffusa* RF2-nonBP9	soil
*B. dolosa* AU0158	CF patient
*B. lata* A05	blood of patient with cepacia syndrome
*B. latens* AU17928	CF Maxillary Sinus
*B. gladioli* ATCC10248	plant
*B. glumae* 257SH-1	rice panicles
*B. metallica* FL-6-5-30-S1-D7	soil
*B. multivorans* ATCCBAA-247	CF patient
*B. pyrrocinia* DSM10685	soil
*B. seminalis* FL-5-4-10S1-D7	soil
*B. stabilis* ATCC BAA-67	CF sputum
*B. stagnalis* MSMB735WGS	soil
*B. territorii* RF8-non-BP5	soil
*B. ubonensis* MSMB1471WGS	soil
*B. vietnamiensis* AU1233	CF blood

**Table 2 vaccines-11-01039-t002:** Isoelectric focusing program.

Time	Voltage (V)
1 h	0
8 h	30
1 h	12
30 min	300
3 h	Up to 3500
10 min	500
Overnight	7950

**Table 3 vaccines-11-01039-t003:** Selected antigen candidates.

Protein Name	Functional Prediction	Protein Type	Length aa	Cellular Localization Prediction	N° of Predicted TMD	VaxiJen Score	Vaxign-ML Score
BCAL0151	ABC-type branched-chain amino acid transport systems, periplasmic component	Extracellular ligand binding protein	381	unknown	Signal peptide	0.5863	98.9
BCAL0198	Outer membrane protein, OmpW	Putative outer membrane protein	278	unknown	8	0.6727	90.9
BCAL0199	DUF2957 domain-containing protein, putatative lipoprotein	Lipoprotein	414	unknown	Signal peptide	0.6030	95.2
BCAL0200	DUF2957 domain-containing protein, putative lipoprotein	Lipoprotein	477	extracellular	≤1	0.7576	90.9
BCAL0358	peptidase M1 family M1 metalopeptidase	Enzyme	723	extracellular	≤1	0.4835	99.7
BCAL1524	Collagen-like triple helix repeat-containing protein	Lipoprotein	558	extracellular	≤1	0.9966	90.9
BCAL2229	Beta-propeller fold lactonase family protein, surface antigen	Enzyme	330	extracellular	≤1	0.3297	90.9
BCAL2615	Putative exported outer membrane porin protein	Putative outer membrane protein	359	outermembrane	16	0.6883	91.0
BCAL3279	DUF3971 domain-containing protein, Possible exported protein	Hypothetical protein	1400	extracellular	1	0.5901	90.9
BCAL3353	Putative outer membrane autotransporter	Autotransporter/adhesin protein	1772	outermembrane/extracellular	≤1	0.8135	94.9
BCAM0949	LipA triacylglycerol lipase	Enzyme	365	extracellular	1	0.4937	90.9
BCAM1514	Outer membrane protein	Other protein	294	outermembrane	≥1	0.6875	90.9
BCAM1737	Alpha-2-macroglobulin	Other protein	2021	outermembrane	1	0.6400	97.7
BCAM1740	Adhesin	Lipoprotein	233	extracellular	≥1	0.6559	90.9
BCAM1931	Outer membrane porin	Putative outer membrane protein	360	outermembrane	16	0.6868	92.5
BCAM2311	Outer membrane porin OmpC	Putative outer membrane protein	380	outermembrane	16	0.5846	91.4
BCAM2328	Coagulation factor 5/8 type-like protein	Other protein	470	extracellular	1	0.5850	95.4
BCAM2418	Cell surface protein putative haemagglutinin-related autotransporter/adhesin protein	Autotransporter/ adhesin protein	558	extracellular	1	0.9141	93.5
BCAM2444	NHL-superfamily, Six-bladed beta-propeller, TolB-like	Other protein	643	extracellular	≤1	0.5015	95.0
BCAS0147	YncE super family, beta-propeller fold lactonase family protein	Enzyme	397	outermembrane/extracellular	1	0.6027	90.9
BCAS0236	putative haemagglutinin-related autotransporter/adhesin protein	Autotransporter/ adhesin protein	1497	outermembrane/extracellular	≤1	0.8682	92.2
BCAS0321	Autotransporter	Autotransporter/ adhesin protein	4250	outermembrane	≥1	1.115	90.9
BCAS0335	putative haemagglutinin-related autotransporter/adhesin protein	Autotransporter/ adhesin protein	1198	extracellular	≤1	0.8436	98.9
BCAS0409	M4 family metallopeptidase	Enzyme	566	extracellular	≤1	0.6513	95.0
BCAS0641	phosphatase PAP2 family protein	Enzyme	464	unknown	≤1	0.6073	93.9

**Table 4 vaccines-11-01039-t004:** Antimicrobial susceptibilities (μg/mL) of *B. cenocepacia* mutants and of the complemented strains.

	Antibiotics									
Strain	AMK	AZT	CIP	LVX	MEM	MIN	NAL	PIP	SPX	TOB
K56-2	≥256	≥256	2	4	8	8	16	128	4	≥256
∆BCAL1524	≥256	256	2	4	8	4	8	128	4	≥256
∆BCAM0949	≥256	256	2	4	4	16	16	**32**	4	≥256
∆BCAS0335	≥256	256	4	4	8	**64**	16	128	4	≥256
∆BCAL1524 pSCrhaB2*BCAM1524*	≥256	≥256	≤2	4	16	≤2	16	64	4	≥256
∆BCAM0949 pSCrhaB2*BCAM0949*	≥256	≥256	≤2	4	4	≤2	4	**128**	4	≥256
∆BCAS0335 pSCrhaB2*BCAS0335*	≥256	≥256	≤2	4	8	**4**	8	128	4	≥256

AMK, amikacin; ATM, aztreonam; CIP, ciprofloxacin; LVX, levofloxacin; MEM, meropenem; MIN, minocycline; NAL, nalidixic acid; PIP, piperacillin; SPX, sparfloxacin; TOB, tobramycin.

## Data Availability

The datasets generated and analyzed during the current study are available from the corresponding author on reasonable request.

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
