# Peer review of "Identification by Reverse Vaccinology of Three Virulence Factors in Burkholderia cenocepacia That May Represent Ideal Vaccine Antigens"

_vaccines, 2023, doi:10.3390/vaccines11061039_

Round 1

Reviewer 1 Report

Dear Authors;

I reviewed your paper entitling: “Characterization of three antigen candidates identified by reverse vaccinology in Burkholderia cenocepacia”. I think the obtained data would have implications clinically for the treatment of CF patients, after in vivo studies. And, I found it very interesting with the scientific results for publication.

Best Regards

Please find my answers to your questions concerning this manuscript:

1. What is the main question addressed by the research?

- Whether or not Reverse vaccinology can be used to identify antigens from a short-list of 24 proteins in Burkholderia cenocepacia candidates as vaccine.

2. Do you consider the topic original or relevant in the field? Does it address a specific gap in the field?

- In my opinion, the topics is relevant in the field and the authors used a combined methods of bioinformatics, biotechnology, proteomics and reverse vaccinology to design a potent vaccine against B. cenocepacia.

3. What does it add to the subject area compared with other published material?

- I think from the proteome of bacteria they have succeeded to identify three specific antigens that are localized in the outer membrane vesicles that are exposed to immune cells, which are not studied in other related published papers in my searches.

4. What specific improvements should the authors consider regarding the

methodology? What further controls should be considered?

- It is good enough and no need to add further experimental methods.

5. Are the conclusions consistent with the evidence and arguments presented

and do they address the main question posed?

- My answer is yes, and there is no contradiction.

6. Are the references appropriate?

- yes

7. Please include any additional comments on the tables and figures.

- Perfect and no additional comments.

Author Response

We thank the reviewer for the kind comments

Reviewer 2 Report

The manuscript entitled "Characterization of three antigen candidates identified by reverse vaccinology in Burkholderia cenocepacia" reports in silico selection of potential virulence factors of B. cenocepacia, and characterization of three proteins via deletion mutants. The manuscript was presented as if the potential antigens were characterized. However, the research was designed to identify the virulence factors. The antigenicity of the proteins should be shown at least in model animals to talk about the potential antigens. Therefore, either the manuscript should be presented as "virulence factors" or immunogenicity experiments should be performed to address the "antigens". The conclusion should be revised accordingly.   

- There is an unnecessary use of capital letters, e.g. "Cystic Fibrosis", "Multi-Drug Resistance", "Outer Membrane Vesicles". 

- "Galleria mellonella" should not be written italic if the title was written italic.

- The full (genus) names of the bacteria should only be written in the first mention, and only the first letter in the next ones. 

Author Response

The manuscript entitled "Characterization of three antigen candidates identified by reverse vaccinology in Burkholderia cenocepacia" reports in silico selection of potential virulence factors of B. cenocepacia, and characterization of three proteins via deletion mutants. The manuscript was presented as if the potential antigens were characterized. However, the research was designed to identify the virulence factors. The antigenicity of the proteins should be shown at least in model animals to talk about the potential antigens. Therefore, either the manuscript should be presented as "virulence factors" or immunogenicity experiments should be performed to address the "antigens". The conclusion should be revised accordingly.   

Authors: We thank the referee for the comments. We revised the conclusion according to the referee's suggestions. We underlined that our work aims to be a proof of concept of “reverse vaccinology” applied on Burkholderia species and that proteins identified have to be considered “antigens candidates” and immunogenicity experiments will be essential to determine whether these proteins are true antigens or not. Moreover, we underlined the connection between virulence factors and antigen candidates to explain why we focused our attention on the role of these proteins in virulence pathways.

We also changed the title of the manuscript to: “Identification by reverse vaccinology of three virulence factors in Burkholderia cenocepacia that may represent ideal vaccine antigens”

- There is an unnecessary use of capital letters, e.g. "Cystic Fibrosis", "Multi-Drug Resistance", "Outer Membrane Vesicles". 

Authors: Unnecessary capital letters were removed.

- "Galleria mellonella" should not be written italic if the title was written italic.

Authors: “Galleria mellonella” was corrected in the title.

- The full (genus) names of the bacteria should only be written in the first mention, and only the first letter in the next ones. 

Authors: Bacteria names were corrected.

Reviewer 3 Report

The manuscript is well writen and the conclusion is fully supported by experimental results. Some minor points need to be clarified before publication.

- Complementation assay in figure 8 (C & D) requires the entire BCAM0949-BCAM0950 operon since the lipase (BCAM0949) must be co-expressed with the foldase (BCAM0950) which means the BCAM0949 cannot fold properly by itself. However, the same protein (BCAM0949) was produced independently for assays in figures 10 & 11. Authors should explain why expression of the lipase (BCAM0949) sometime requires BCAM0950 and sometime not.

- Health index score of the wild type (K56-2) in figure 9 is quite different between figure A & C. Why?

- Page 17 line 9, BCAM949 should be BCAM0949

- Page 17, 3rd-line in the second paragraph "24 post-infection" mean?

Author Response

The manuscript is well writen and the conclusion is fully supported by experimental results. Some minor points need to be clarified before publication.

- Complementation assay in figure 8 (C & D) requires the entire BCAM0949-BCAM0950 operon since the lipase (BCAM0949) must be co-expressed with the foldase (BCAM0950) which means the BCAM0949 cannot fold properly by itself. However, the same protein (BCAM0949) was produced independently for assays in figures 10 & 11. Authors should explain why expression of the lipase (BCAM0949) sometime requires BCAM0950 and sometime not.

Authors: We thank the reviewer for the question. Papadopoulos et al., [68] and Putra et al., [69] demonstrated that P. aeruginosa LipA and B. territori LipBT lipases have to be co-expressed with the relative foldase to avoid precipitation and to allow increased functional protein concentration, which would be required to complement some phenotypes, suggesting that complementation depends on lipase concentration.  While in the case of the lipolytic activity of the protein and rhamnolipid production (Fig. 10 and 11) the BCAM0949 gene itself was enough to get the complemented phenotype, this was not true for the swimming motility experiments (Fig. 8), indicating that this is one of the phenotypes for which a higher concentration of functional protein is required for the complementation.

- Health index score of the wild type (K56-2) in figure 9 is quite different between figure A & C. Why?

Authors: The health index scores were different because in the first experiment (Figure 9A) the negative control is represented by G. mellonella infected  with the K56-2 strain. In Figure 9C the control was represented by G. melonella infected with the K56-2 carrying the empty vector pAP20 used for the complementation, which gives a different result.

- Page 17 line 9, BCAM949 should be BCAM0949

Authors: The taping error was corrected.

- Page 17, 3rd-line in the second paragraph “24 post-infection” mean?

Authors: We corrected the sentence. It refers to 24 hours post-infection.

Reviewer 4 Report

In the manuscript, the authors exploited for the first time “reverse vaccinology” approach to Burkholderia cenocepacia to identify novel antigens' candidates. Since B. cenocepacia is one of the leading pathogen in CF patients and the vaccine against that pathogen does not exist, the research topic addressed is very important and there is very useful data presented. Experimental and statistical methods are fine. Conclusions are conducted in a balanced way. It is commendable that the functionality of three studied surface proteins was confirmed not only by comparing mutants to controls, but also using complementary strains. This is not a common practice today. The manuscript is  well written but there is one part of the manuscript – the discussion  – that need a major rewriting to improve clarity.

Title and Introduction

    In my opinion the title of this paper could be more informative and could contain also virulence and phenotypic aspects of the data. It is important, as the present title is somewhat misleading suggesting the immunogenic studies, which are not the goal of this paper. That is my suggestion.  

2.      Is the citation 27 (from 2000)  properly placed, if this study identify novel B. cenocepacia antigens?

Methods

1.      There is no information about discarding flagella while obtaining OMVs. Was such a procedure followed? Since the flagella may cause the contamination of OMVs preparation, this issue need to be  clarified.

2.      Actually, I do not understand the statement: …..proteins with an identity of 100% in all pathogenic strains were excluded, as they are probably not exposed to the immune system….(page: 3). Why would the immunogenicity of proteins depend on their degree of identity? Please explain?.

Results

1.      Why Kaplan-Meyer survival curves for  Galleria mellonella are not  included in the study? They would be very useful.

2.      Page 17: According to Figure 9 B it seems that mutant DBCAM0949 caused killing of the larvae twice higher than WT strain K56-2 – the text says that it did not affect larval survival in comparison with WT. If that is the result of only one experiment, the survival curves  of the whole set of data should be added to dispel doubts.

3.      The quality and resolution of Figure 4 as well as Figure 6a should be improved.

Discussion

Discussion is the weakest part of the paper. The authors only summarize the results and do not discuss them against other studies. This part should be rewritten.

Author Response

In the manuscript, the authors exploited for the first time “reverse vaccinology” approach to Burkholderia cenocepacia to identify novel antigens' candidates. Since B. cenocepacia is one of the leading pathogen in CF patients and the vaccine against that pathogen does not exist, the research topic addressed is very important and there is very useful data presented. Experimental and statistical methods are fine. Conclusions are conducted in a balanced way. It is commendable that the functionality of three studied surface proteins was confirmed not only by comparing mutants to controls, but also using complementary strains. This is not a common practice today. The manuscript is  well written but there is one part of the manuscript – the discussion  – that need a major rewriting to improve clarity.

Authors: We thank the referee for the comments. We revised the manuscript to improve clarity and addressed the main referee’s issues.

Title and Introduction

    In my opinion the title of this paper could be more informative and could contain also virulence and phenotypic aspects of the data. It is important, as the present title is somewhat misleading suggesting the immunogenic studies, which are not the goal of this paper. That is my suggestion. 

Authors: We thank the referee for the comments. The title has been changed to ”Identification by reverse vaccinology of three virulence factors in Burkholderia cenocepacia that may represent ideal vaccine antigens”

  1. Is the citation 27 (from 2000) properly placed, if this study identify novel B. cenocepacia antigens?

Authors: The citation refers to the “reverse vaccinology” method, so it has been placed before in the sentence. Now it reads: “In this article we exploited for the first time the “reverse vaccinology” approach [27] to B. cenocepacia to identify novel  antigen candidates.”

Methods

  1. There is no information about discarding flagella while obtaining OMVs. Was such a procedure followed? Since the flagella may cause the contamination of OMVs preparation, this issue need to be clarified.

Authors: The description of the protocol to isolate OMV was modified in the text adding a reference (Klimentova and Stulik, 2015). Ultracentrifugation with 100kDa filter allowed removal of all non-OMV associated proteins, avoiding contamination of OMV.

  1. Actually, I do not understand the statement: …..proteins with an identity of 100% in all pathogenic strains were excluded, as they are probably not exposed to the immune system….(page: 3). Why would the immunogenicity of proteins depend on their degree of identity? Please explain?.

Authors: We thank the referee for the comment. Proteins displaying 100% identity in all the strains belonging to the Bcc are not affected by the immunity selective pressure, suggesting a low immunogenicity during infection (Sette and Rappuoli, 2010). On the contrary, a good antigen candidate has to be highly immunogenic.  The relative sentence and reference were added in the text.

Results

  1. Why Kaplan-Meyer survival curves for Galleria mellonella are not  included in the study? They would be very useful.
  2. Page 17: According to Figure 9 B it seems that mutant DBCAM0949 caused killing of the larvae twice higher than WT strain K56-2 – the text says that it did not affect larval survival in comparison with WT. If that is the result of only one experiment, the survival curves of the whole set of data should be added to dispel doubts.

Authors: We thank the referee for the comments. In our opinion the health-index-score, that takes into account the movement, the cocoon formation, the melanization and the survival of the larvae is more precise and gives more information to determine the effect of the infections (Tsai et al., 2016), compared to the only Kaplan-Meyer survival curves.

The ΔBCAM0949 caused a higher melanization score (shown in Figure 9B) but the larvae were still alive. Moreover the difference in the health index score between the larvae injected with B. cenocepacia WT or with ΔBCAM0949 was not statistically significant (Figure 9A). All the presented data are the mean of the results from three independent experimental replicates.

  1. The quality and resolution of Figure 4 as well as Figure 6a should be improved.

Authors: The resolutions of Figure 4 and 6a were checked and were prepared according to the journal Instructions for Authors.

Discussion

Discussion is the weakest part of the paper. The authors only summarize the results and do not discuss them against other studies. This part should be rewritten.

Authors: As allowed in the journal Instructions for Authors, we decided to combine the “Results and Discussion” sections in order to allow a point-by-point interpretation of each result in perspective of previous studies and of the working hypotheses. The final, optional “Conclusions” section, was added to summarize the main findings and impact of our research.

Indeed, the discussion against other studies has been reported in each paragraph of “Results and Discussion”. A few examples are reported below:

3.1. In silico identification of antigen candidates: here we describe and discuss why the three proteins characterized in the manuscript were selected based on previous studies.

“Three antigens, the collage-like protein BCAL1524, the lipase LipA BCAM0949 and the autotransporter adhesin BCAS0335 were selected for further analysis.  The rationale for the selection is described below. Collagen-like proteins, characterized by a collagen-like (CL) domain containing Gly-Xaa-Yaa amino acid repetition and organized in a triple helix-structure, are widespread in pathogenic bacteria and highly resemble human collagen [58]. Due to their structure, they can interact with different host factors promoting adhesion, inflammation and immunoreaction [59,60]. Additionally, Grund et al., [61] described the high Th2 immune response induced in mice immunized with B. pseudomallei collagen-like protein 8 (Bucl8) antigens, while studies on the role of these proteins in B. cenocepacia are still lacking. Lipases are commonly found in clinical B. cepacia complex isolates [62]. While evidence of their involvement in Bcc infection is still little [62,63], multiple studies reported their role in virulence and pathogenesis in P. aeruginosa [64,65,51]. Trimeric autotransporter adhesins are known to play a key role in virulence for a wide range of bacteria [66]; Pimenta et al., [25] and Mil-Homens et al., [26,67] extensively described the role in adhesion and inflammation in epidemic B. cenocepacia epidemic strain K56-2, thus highlighting the putative role they may play as protective antigens.”

3.3. Analysis of protein localization

“An important aspect of antigen candidates is their localization. To support the in silico prediction, a proteomic analysis of the deleted strains was performed and demonstrated that the three proteins are indeed localized in the outer membrane vesicle (OMV) compartments. The OMVs play a critical role in host-pathogen interaction and virulence, as they are produced in response to stress conditions and can carry bioactive molecules and virulence factors able to sustain bacterial growth and modulate host inflammation [71]; hence therefore the presence of the three proteins in these compartments highlights their possible role in these pathways. Moreover, their localization on the bacterial surface could make them ideal targets for the recognition by the host immune system.”

3.7. Characterization of the lipase BCAM0949

“Although the deletion of BCAM0949 has no effect on the in vivo virulence of Burkholderia, its amino acid sequence is characterized by a conserved domain of the triacylglycerol esterase/lipase superfamily, so we further characterized its activity. In other bacteria, such as Pseudomonas aeruginosa, several lipolytic enzymes are secreted or surface-exposed, among these EstA which has a lipolytic activity, and are involved in rhamnolipid production and virulence [52].”

Round 2

Reviewer 2 Report

The concerns of this reviewer have been addressed.

The authors should correct typos.